# Arc-Induced Long-Period Fiber Gratings at INESC TEC. Part I: Fabrication, Characterization and Mechanisms of Formation

**DOI:** 10.3390/s21144914

**Published:** 2021-07-19

**Authors:** Gaspar Rego, Paulo Caldas, Oleg V. Ivanov

**Affiliations:** 1proMetheus, Instituto Politécnico de Viana do Castelo, Rua Escola Industrial e Comercial Nun’Álvares, 4900-347 Viana do Castelo, Portugal; pcaldas@estg.ipvc.pt; 2Center for Applied Photonics, INESC TEC, Rua Dr. Roberto Frias, 4200-465 Porto, Portugal; 3Ulyanovsk Branch of Kotel’nikov Institute of Radio Engineering and Electronics of Russian Academy of Sciences, Ulitsa Goncharova 48, 432071 Ulyanovsk, Russia; olegivvit@yandex.ru

**Keywords:** long-period fiber grating, electric arc technique, mechanisms of gratings formation

## Abstract

In this work, we reviewed the most important achievements of INESC TEC related to the fabrication of long-period fiber gratings using the electric arc technique. We focused on the fabrication setup, the type of fiber used, and the effect of the fabrication parameters on the gratings’ transmission spectra. The theory was presented, as well as a discussion on the mechanisms responsible for the formation of the gratings, supported by the measurement of the temperature reached by the fiber during an electric arc discharge.

## 1. Introduction

Long-period fiber gratings (LPFGs) are passive optical-fiber components that result from periodic modulations of the fiber core with a periodicity ranging from 100 μm to 1 mm [1]. The periodic perturbations enable the coupling between the fundamental core mode and several copropagating cladding modes. Thus, the grating transmission spectrum exhibits several dips corresponding to those couplings at discrete wavelengths (wavelength selective optical filters). The resonance wavelengths and the amplitude of the dips depend on physical parameters such as temperature, strain, bending, torsion, pressure, and external refractive index and, therefore, they can be used in optical communications and sensing applications [2,3,4,5,6,7,8,9,10,11,12,13,14,15,16,17,18,19,20,21,22,23,24,25,26,27,28,29,30,31,32,33,34,35,36]. Among the different techniques available to produce LPFGs [37,38,39,40,41,42,43,44,45,46,47,48,49,50,51,52,53,54,55,56,57,58,59,60,61,62,63], the electric arc technique is one of the most used for the following reasons: it is low cost, flexible, reproducible, and can virtually induce gratings in any kind of fiber [64,65,66,67,68,69,70,71,72,73,74,75,76,77,78,79]. Therefore, it does not require the fiber to be photosensitive, or to be hydrogen treated followed by thermal annealing, as is the case for the ones induced by UV-laser radiation. Furthermore, arc-induced gratings can tolerate very high temperatures, a property that is also shared by LPFGs fabricated by CO_2_ laser radiation and infrared femtosecond laser radiation [49,64,80,81], being useful for temperature monitoring at extreme environments [82]. The most well-known application of the electric arc technique is fiber fusion splicing [83]; however, there are a number of applications besides LPFGs fabrication, such as fiber tapering, fiber couplers, microspheres, etc. [84]. The first reported fabrication of LPFGs may be attributed to Poole et al. in 1994, although they used a two-step process involving ablation of the cladding of a two-mode fiber by CO_2_ laser radiation followed by annealing through arc discharges [85]. In 1997, Dianov et al. demonstrated the fabrication of LPFGs by using only arc-discharges [38]. Since then, the technique spread worldwide and several hundreds of papers have been published [86]. In the following sections we discuss the state-of-the-art achievements of INESC TEC related to fabrication setup, to the fabrication parameters and the type of fiber used, and their influence on the gratings’ transmission spectra. We also present the theoretical aspects and discuss the mechanisms of gratings formation supported by the measurement of the temperature reached by the fiber during an arc discharge.

## 2. Theory of Long-Period Fiber Gratings

The simulation of spectra of an LPFG has two main steps: calculation of modes (propagation constants, mode fields) and solving how modes propagate through the fiber (using coupled mode equations or the transfer matrix method). For optical fibers with step index profiles, exact analytical expressions for the modes fields can be obtained. The propagation constants of modes can be found numerically from the exact dispersion equation [87]. In the elementary statement of the problem of propagation of cladding modes in optical fibers, a structure should consist of a minimum of three layers: core, cladding, and ambient medium. The solution of this problem was obtained in an explicit form in [88] on the basis of the general method for an arbitrary number of layers [87]. In this section, we present a matrix method of calculation of cladding modes of a multilayer cylinder optical fiber with an arbitrary number of layers.

Let us consider an optical fiber with an azimuthally symmetric step-function profile of dielectric permittivity ε(r). Electric and magnetic fields of a mode of the cylindrical waveguide have the following form:(1)$$\begin{array}{l} E(r,\varphi ,z,t)=E(r)\exp[i(\beta z-\omega t+\nu \varphi )],\\ H(r,\varphi ,z,t)=H(r)\exp[i(\beta z-\omega t+\nu \varphi )], \end{array}$$
where β is the propagation constant, ω is the circular frequency, and ν is the azimuthal mode number. From the Maxwell equations written in the cylindrical coordinates, it is possible to obtain the following scalar equations for the longitudinal components of electric and magnetic fields:(2)$$\begin{array}{l} \frac{d^{2}E^{\left(z\right)}}{dr^{2}}+\frac{1}{r}\frac{dE^{\left(z\right)}}{dr}+\left(k_{0}^{2}\varepsilon (r)-\beta_{}^{2}-\nu^{2}/r^{2}\right)E^{\left(z\right)}=0,\\ \frac{d^{2}H^{\left(z\right)}}{dr^{2}}+\frac{1}{r}\frac{dH^{\left(z\right)}}{dr}+\left(k_{0}^{2}\varepsilon (r)-\beta_{}^{2}-\nu^{2}/r^{2}\right)H^{\left(z\right)}=0, \end{array}$$
where $k_{0}=\omega /c$ is the wavenumber and *c* is the speed of light in a vacuum. The solution of differential Equation (2) in the general case is expressed through the Bessel functions:(3)$$\begin{array}{l} E^{\left(z\right)}=i\frac{u^{2}}{k_{0}\varepsilon }\left(AJ_{\nu }(ur)+BY_{\nu }(ur)\right),\\ H^{\left(z\right)}=-\frac{u^{2}}{k_{0}}\left(CJ_{\nu }(ur)+DY_{\nu }(ur)\right), \end{array}$$
where $u=\sqrt{k_{0}^{2}\varepsilon -\beta^{2}}$ is the transverse component of the wavevector, which is a real quantity for the fields oscillating along the radius and is an imaginary quantity for the exponentially decreasing fields. The field components $E^{\left(\varphi \right)}$ and $H^{\left(\varphi \right)}$, which are tangential to the boundaries of layers of the waveguide, can be found from $E^{\left(z\right)}$ and $H^{\left(z\right)}$ and are represented as follows:(4)$$\begin{array}{l} E^{\left(\varphi \right)}=-\frac{i\sigma }{\varepsilon r}\left[AJ_{\nu }(ur)+BY_{\nu }(ur)\right]+iu\left[C{J^{\prime }}_{\nu }(ur)+D{Y^{\prime }}_{\nu }(ur)\right],\\ H^{\left(\varphi \right)}=-u\left[AJ_{\nu }(ur)+BY_{\nu }(ur)\right]+\frac{\sigma }{r}\left[C{J^{\prime }}_{\nu }(ur)+D{Y^{\prime }}_{\nu }(ur)\right], \end{array}$$
where $\sigma =\beta \nu /k_{0}$.

The radial components are also expressed through $E^{\left(z\right)}$ and $H^{\left(z\right)}$ and have the form:(5)$$\begin{array}{l} E^{\left(r\right)}=-\frac{\beta u}{k_{0}\varepsilon }\left[A{J^{\prime }}_{\nu }(ur)+B{Y^{\prime }}_{\nu }(ur)\right]+\frac{\nu }{r}\left[CJ_{\nu }(ur)+DY_{\nu }(ur)\right],\\ H^{\left(r\right)}=i\frac{\nu }{r}\left[AJ_{\nu }(ur)+BY_{\nu }(ur)\right]-\frac{i\beta u}{k_{0}}\left[C{J^{\prime }}_{\nu }(ur)+D{Y^{\prime }}_{\nu }(ur)\right]. \end{array}$$

Thus, the fields $E^{\left(r\right)}$, $H^{\left(z\right)}$, and $H^{\left(\varphi \right)}$ are real, and $E^{\left(\varphi \right)}$, $E^{\left(z\right)}$, and $H^{\left(r\right)}$ are purely imaginary. In order to find the solution of the Maxwell’s equations in the case of a multilayer cylindrical waveguide, it is necessary to use the boundary conditions for the tangential components of the fields, the condition of finiteness of field amplitudes at the fiber center, and the condition of zero field amplitude at infinity. Further, we use indexed variables $A_{i}$, $B_{i}$, $C_{i}$, $D_{i}$, $u_{i}$, and $r_{i}$ for a particular cylindrical layer having the index *i*.

From the condition of finiteness of the fields at the point r=0, it follows that $B_{1}$ and $D_{1}$ for the central cylinder should be equal to zero. Then fields inside this cylinder that are tangential to its boundaries can be written in a matrix form:(6)$$\left(\begin{array}{c} ik_{0}E^{\left(z\right)}\\ -k_{0}H^{\left(z\right)}\\ iE^{\left(\varphi \right)}\\ H^{\left(\varphi \right)} \end{array}\right)=\left(\begin{array}{cccc} -(u_{1}^{2}/\varepsilon_{1})J_{\nu }(u_{1}r) & 0 & 0 & 0\\ 0 & 0 & u_{1}^{2}J_{\nu }(u_{1}r) & 0\\ \left(\sigma /r\varepsilon_{1})J_{\nu }(u_{1}r\right) & 0 & -u_{1}^{}{J^{\prime }}_{\nu }(u_{1}r) & 0\\ -u_{1}{J^{\prime }}_{\nu }(u_{1}r) & 0 & \left(\sigma /r)J_{\nu }(u_{1}r\right) & 0 \end{array}\right)\left(\begin{array}{c} A_{1}\\ 0\\ C_{1}\\ 0 \end{array}\right)=M_{1}\left(\begin{array}{c} A_{1}\\ 0\\ C_{1}\\ 0 \end{array}\right).$$In the subsequent cylindrical layers, the tangential fields have the form:(7)$${\left(ik_{0}E^{\left(z\right)},\;-k_{0}H^{\left(z\right)},\;iE^{\left(\varphi \right)},\;H^{\left(\varphi \right)}\right)}^{T}=M_{i}{\left(A_{i},\;B_{i},\;C_{i},\;D_{i}\right)}^{T}$$
where
(8)$$M_{i}=\left(\begin{array}{cccc} -(u_{i}^{2}/\varepsilon_{i})J_{\nu }(u_{i}r) & -(u_{i}^{2}/\varepsilon_{i})Y_{\nu }(u_{i}r) & 0 & 0\\ 0 & 0 & u_{i}^{2}J_{\nu }(u_{i}r) & u_{i}^{2}Y_{\nu }(u_{i}r)\\ \left(\sigma /r\varepsilon_{i})J_{\nu }(u_{i}r\right) & \left(\sigma /r\varepsilon_{i})Y_{\nu }(u_{i}r\right) & -u_{i}^{}{J^{\prime }}_{\nu }(u_{i}r) & -u_{i}^{}{Y^{\prime }}_{\nu }(u_{i}r)\\ -u_{i}{J^{\prime }}_{\nu }(u_{i}r) & -u_{i}{Y^{\prime }}_{\nu }(u_{i}r) & \left(\sigma /r)J_{\nu }(u_{i}r\right) & \left(\sigma /r)Y_{\nu }(u_{i}r\right) \end{array}\right).$$

From the condition of zero field amplitude at infinity, it follows that the value of $k_{0}^{2}\varepsilon_{a}-\beta^{2}$, where $\varepsilon_{a}$ is the dielectric permittivity of the ambient medium for the fiber, should be less than zero, and the variables $A_{a}$ and $C_{a}$ should be equal to zero. Then, we have for the ambient layer that
(9)$${\left(ik_{0}E^{\left(z\right)},\;-k_{0}H^{\left(z\right)},\;iE^{\left(\varphi \right)},\;H^{\left(\varphi \right)}\right)}^{T}=M_{a}{\left(0,\;B_{a},\;0,\;D_{a}\right)}^{T},$$
where
$$M_{a}=\left(\begin{array}{cccc} 0 & \left(w_{a}^{2}/\varepsilon_{a})K_{\nu }(w_{a}r\right) & 0 & 0\\ 0 & 0 & 0 & -w_{a}^{2}K_{\nu }(w_{a}r)\\ 0 & \left(\sigma /r\varepsilon_{a})K_{\nu }(w_{a}r\right) & 0 & -w_{a}^{}{K^{\prime }}_{\nu }(w_{a}r)\\ 0 & -w_{a}{K^{\prime }}_{\nu }(w_{a}r) & 0 & \left(\sigma /r)K_{\nu }(w_{a}r\right) \end{array}\right),$$$K_{\nu }$ is the modified Bessel function, and $w_{a}=\sqrt{\beta^{2}-k_{0}^{2}\varepsilon_{a}}$. The boundary conditions are expressed in a matrix form as
(10)$$\begin{array}{c} M_{1}(r_{1}){\left(A_{1},\;0,\;C_{1},\;0\right)}^{T}=M_{2}(r_{1}){\left(A_{2},\;B_{2},\;C_{2},\;D_{2}\right)}^{T},\\ \ldots \\ M_{i}(r_{i}){\left(A_{i},\;B_{i},\;C_{i},\;D_{i}\right)}^{T}=M_{i+1}(r_{i}){\left(A_{i+1},\;B_{i+1},\;C_{i+1},\;D_{i+1}\right)}^{T},\\ \ldots \\ M_{a-1}(r_{a-1}){\left(A_{a-1},\;B_{a-1},\;C_{a-1},\;D_{a-1}\right)}^{T}=M_{a}(r_{a-1}){\left(0,\;B_{a},\;0,\;D_{a}\right)}^{T}. \end{array}$$These equations can be transformed to one matrix equation:(11)$$\begin{array}{c} M_{1}(r_{1}){\left(A_{1},\;0,\;C_{1},\;0\right)}^{T}=M{\left(0,\;B_{a},\;0,\;D_{a}\right)}^{T},\\ M=M_{2}(r_{1})M_{2}^{-1}(r_{2})M_{3}(r_{2})\ldots M_{a-1}^{-1}(r_{a-1})M_{a}(r_{a-1}). \end{array}$$Using the components $m_{ij}$ and $m_{ij}^{\left(1\right)}$ of the matrices M and $M_{1}$, respectively, the relation (11) can be rewritten as
(12)$$N{\left(A_{1},\;B_{a},\;C_{1},\;D_{a}\right)}^{T}=0,\;N=\left(\begin{array}{cccc} -m_{11}^{\left(1\right)} & m_{12}^{} & -m_{13}^{\left(1\right)} & m_{14}^{}\\ -m_{21}^{\left(1\right)} & m_{22}^{} & -m_{23}^{\left(1\right)} & m_{24}^{}\\ -m_{31}^{\left(1\right)} & m_{32}^{} & -m_{33}^{\left(1\right)} & m_{34}^{}\\ -m_{41}^{\left(1\right)} & m_{42}^{} & -m_{43}^{\left(1\right)} & m_{44}^{} \end{array}\right).$$Considering that the determinant of the matrix *N* is equal to zero, we obtain an equation with one unknown β:(13)detN=0,
whose solution is a set of propagation constants of modes. After the values of β are found, they can be substituted to (12), and it becomes possible to find $A_{1}$, $B_{a}$, $C_{1}$, and $D_{a}$ accurate to within an arbitrary common factor and then, with the help of (10), all remaining $A_{i}$, $B_{i}$, $C_{i}$, and $D_{i}$ can be found for the intermediate layers. The fields of modes can be obtained using relations (6)–(9).

The procedure described can be used directly for writing a computer program that solves numerically the problem of finding propagation constants and mode-field profiles of optical fibers. The convenience of this method is that the elements of all matrices and the variables $A_{i}$, $B_{i}$, $C_{i}$, and $D_{i}$ are real quantities.

For the calculation of the core modes, it is necessary to replace the oscillating Bessel functions $J_{\nu }$ and $Y_{\nu }$ in the expression for the matrix describing the field in the cladding by the modified functions $I_{\nu }$ and $K_{\nu }$, respectively, and $u_{i}$ by $w_{i}$. For a cladding with a sufficiently large radius, the field amplitude of the core mode at the outer cladding radius is vanishingly small; therefore, the presence of the ambient medium outside the fiber cladding is usually neglected in the calculation of the core modes.

In spite of a considerable difference between the refractive indices (RIs) of the cladding and ambient medium, in many cases the weak guidance approximation can be used with a rather good accuracy. The advantage of this approximation is that it can be easily applied to optical fibers with arbitrary gradient RI profiles, for example, for dispersion-shifted optical fibers having triangular RI profiles. The weak guidance approximation is also distinguished by its simplicity, as it operates with transversally linearly polarized modes. The linearly polarized modes can be obtained as an exact superposition of two, three, or four hybrid modes of the zero-order paraxial approximation. In the general case, a linearly polarized mode is formed as the sum of two right- and two left-circularly polarized hybrid modes,
(14)$${\mathrm{LP}}_{\lambda ,m}={\mathrm{HE}}_{\lambda +1,m}+{\mathrm{HE}}_{-(\lambda +1),m}+{\mathrm{EH}}_{\lambda -1,m}+{\mathrm{EH}}_{-(\lambda -1),m}.$$The LP_0*m*_ and LP_1*m*_ modes belong to a special case:(15)$$\begin{array}{c} {\mathrm{LP}}_{0,m}={\mathrm{HE}}_{1,m}+{\mathrm{HE}}_{-1,m},\\ {\mathrm{LP}}_{1,m}^{\left(a\right)}={\mathrm{HE}}_{2,m}+{\mathrm{HE}}_{-2,m}+{\mathrm{TE}}_{0,m},\\ {\mathrm{LP}}_{1,m}^{\left(b\right)}={\mathrm{HE}}_{2,m}+{\mathrm{HE}}_{-2,m}+{\mathrm{TM}}_{0,m}. \end{array}$$

As an example, we demonstrate cladding modes of a three-layer optical fiber calculated using the method given above with the following parameters: $n_{\mathrm{co}}=1.4492$, $n_{\mathrm{cl}}=1.444$, $r_{\mathrm{co}}=4.1$ μm, and $r_{\mathrm{cl}}=62.5$ μm at wavelength λ=1.55 μm.

For a fixed value of the azimuthal mode number, the solution of Equation (13) contains two alternating sets of propagation constants, each of which correspond to two types of modes. So, for ν=0, the odd solutions correspond to the TE modes and the even solutions correspond to the TM modes. For ν≥1, the odd solutions correspond to the HE modes and the even solutions correspond to the EH modes.

The core and cladding modes have different phase velocities of propagation, which are directly related to the mode effective RIs and are defined as the ratio between propagation constants of modes and the wave number in a vacuum: $n_{e\mathrm{ff}}=\beta /k_{0}$. The value of the effective RI of the core mode is between the cladding RI and the core RI. The effective RIs of the cladding modes are below the cladding RI.

The dependences of the effective RIs on wavelength λ are shown in Figure 1 for different types of modes of a single-mode optical fiber with the parameters defined above. The solid curves correspond to the HE_1*m+*1_ modes, the dashed curves correspond to the EH_1*m*_ modes, and the dotted curves correspond to the TM_1*m*_ modes. The numbers near the curves in the right part of the figure indicate the radial modal number. The curve corresponding to the core mode has the highest RI. It becomes closer to $n_{\mathrm{cl}}$ with an increase in wavelength. The curves corresponding to the cladding modes are located below the value $n_{\mathrm{cl}}=1.444$. These curves also go down with an increase in λ, and the slope increases for higher order modes. For low-order modes, the slope of the curves is smaller than the slope for the core mode. For some value of the radial mode number, the slope angle of the effective RI of the cladding mode becomes equal to the slope of the curve corresponding to the core mode.

The fiber becomes multimode at short wavelengths, and it can support more than one mode. This is seen in the figure for λ≤1.3 μm: the effective RI of the TM_01_ cladding mode overcomes the limit $n_{\mathrm{cl}}$ and becomes a core mode.

Because the amplitude of the induced gratings light is of order $10^{-3}$, i.e., it is much less than 1, the theory of coupled modes with slowly varying amplitudes can be used. For analyzing the mode coupling it is necessary to calculate the coupling coefficients, which are expressed through the overlap integrals of modes.

Symmetric LPFGs couple modes with identical azimuthal numbers; therefore, the fundamental core mode with ν=1 can be coupled with the HE_1*m*_ and EH_1*m*_ modes. The coupling coefficient for two modes is determined through the overlap integral:(16)$$K_{pq}=\frac{\omega \varepsilon_{0}}{4}\int_{\infty }E_{p}^{*}(r,\varphi )\Delta \varepsilon E_{q}^{}(r,\varphi )dS,$$
where Δε is the change of dielectric permittivity of the fiber as a result of perturbation. In the calculation of coupling coefficients between weak-guided modes, the longitudinal components of electric fields can be neglected, as they are an order of magnitude smaller than the transverse components.

When a photo-induced LPFG is written in an optical fiber, usually only the RI of the doped fiber core is changed; therefore, Δε differs from zero for $r<r_{\mathrm{co}}$. In this area, the RI can be represented in the following form:(17)$$n(z)=n_{\mathrm{co}}\left\{1+\sigma (z)\left[1+M\cos(2\pi z/\Lambda )\right]\right\},$$
where σ(z) is a slowly varying envelope, *M* is the amplitude of modulation of the induced RI in the grating or grating visibility, and Λ is the grating period. Taking (16) into account, the coupling coefficient can be represented as $K_{pq}=\kappa_{pq}\left[1+M\cos(2\pi z/\Lambda )\right]$, where
(18)$$\kappa_{pq}=\frac{\omega \varepsilon_{0}}{2}n_{\mathrm{co}}^{2}\sigma (z)\int_{0}^{r_{\mathrm{co}}}E_{p}^{*}(r,\varphi )E_{q}^{}(r,\varphi )dS.$$

The coupling with the EH_1*m*_ modes with small radial numbers is very weak, in contrast to the HE_1*m*_ modes, which is related to a small amplitude of the field of these EH_1*m*_ modes near the fiber core.

For antisymmetric LPFGs such as micro-bend and arc-induced, the perturbation Δε is a function of the angle φ:(19)$$n(z)=n_{\mathrm{co}}\left\{1+\sigma (z)M\sin\varphi \cos(2\pi z/\Lambda )\right\},$$

In this case, the coupling coefficients are non-zero between the core HE_1*m*_ mode and HE_2*m*_, TE*_m_*, and TM*_m_* modes. The LP_0*m*_ mode is coupled to LP_1*m*_ modes in the weak-guidance approximation.

There are two main methods for calculating the transmission spectra of LPFGs. The first of them is based on the standard coupled-mode theory, which uses the phase matching approximation and the small perturbation approximation [88,90]. When calculating gratings with a non-sinusoidal and rectangular shape of the RI profile along the fiber, only the zero and first harmonics are taken into account and the contributions of the higher harmonics of the Fourier spectrum of the grating are neglected.

The second method is the transfer matrix method. It is based on dividing the fiber into sections with a uniform cross-section along the fiber [91]. For each section, the mode structure and its transmission matrix are calculated, and the matrix relates the electromagnetic field at the input and output. Multiplication of the matrices for each section results in the matrix describing the grating as a whole. The transfer matrix method provides an exact solution.

Let us consider in more detail the application of the coupled-mode method. We can neglect the interaction between the cladding modes, as the overlap integral is small. We considered only the self-coupling of modes, which shift resonances in the transmission spectra of LPFGs. With these approximations, the coupled-mode equations can be written in the following form [88]:(20)$$\begin{array}{l} \frac{dA^{\mathrm{co}}}{dz}=i\kappa^{\mathrm{co}-\mathrm{co}}A^{\mathrm{co}}+i\sum_{n}\frac{M}{2}\kappa_{n}^{\mathrm{co}-\mathrm{cl}}A_{n}^{\mathrm{cl}}\exp(-i2\delta_{n}^{\mathrm{co}-\mathrm{cl}}z),\\ \frac{dA_{m}^{\mathrm{cl}}}{dz}=i\kappa_{m}^{\mathrm{cl}-\mathrm{cl}}A_{m}^{\mathrm{cl}}+i\frac{M}{2}\kappa_{m}^{\mathrm{cl}-\mathrm{co}}A^{\mathrm{co}}\exp(i2\delta_{m}^{\mathrm{co}-\mathrm{cl}}z), \end{array}$$
where $\delta_{m}^{\mathrm{co}-\mathrm{cl}}=\left(\beta^{\mathrm{co}}-\beta_{m}^{\mathrm{cl}}-2\pi /\Lambda \right)/2$, and rapidly oscillating terms are neglected. The resonance condition has the form
(21)$$\beta^{\mathrm{co}}+\Delta \beta^{\mathrm{co}}-\beta_{m}^{\mathrm{cl}}-\Delta \beta_{m}^{\mathrm{cl}}=2\pi /\Lambda ,$$
where $\Delta \beta^{\mathrm{co}}=\kappa^{\mathrm{co}-\mathrm{co}}$ and $\Delta \beta_{m}^{\mathrm{cl}}=\kappa_{m}^{\mathrm{cl}-\mathrm{cl}}$ are corrections to the propagation constants of the core and cladding modes arising from the self-coupling of the modes at the zero Fourier component of the grating-refractive index.

Usually, cladding-mode resonances are far apart from each other in the spectrum; therefore, in the coupled-mode equations at fixed wavelengths, only one cladding mode can be taken into account. For a grating with a uniform distribution and coupling two fiber modes of the induced refractive index, it is easy to obtain an analytical solution to system (21). Further, we assume that the constant component of σ is equal to zero. In this case, the solution of (21) can be written in the form
(22)$$\left(\begin{array}{c} A^{\mathrm{co}}(L)\\ A_{m}^{\mathrm{cl}}(L) \end{array}\right)=\left(\begin{array}{cc} e^{-i\delta z} & 0\\ 0 & e^{i\delta z} \end{array}\right)\left(\begin{array}{cc} \cos\eta L+i(\delta /\eta )\sin\eta L & i(\kappa /\eta )\sin\eta L\\ i(\kappa^{*}/\eta )\sin\eta L & \cos\eta L-i(\delta /\eta )\sin\eta L \end{array}\right)\left(\begin{array}{c} A^{\mathrm{co}}(0)\\ A_{m}^{\mathrm{cl}}(0) \end{array}\right),$$
where $\eta =\sqrt{\delta^{2}+{\left|\kappa \right|}^{2}}$, $\kappa =\kappa_{n}^{\mathrm{co}-\mathrm{cl}}M/2$, and the indices of the detuning parameter have been omitted here to be succinct.

Assuming that the calculation of the mode structure of the fiber is performed considering the constant RI of the fiber, the corresponding self-coupling corrections to the propagation constants can be equated to zero. In this case, the resonance wavelengths are determined by the relation
(23)$$\lambda =\left(n_{\mathrm{eff}}^{\mathrm{co}}-n_{\mathrm{eff},m}^{\mathrm{cl}}\right)\Lambda .$$

When measuring the transmission spectrum, the initial condition for the cladding mode is that its amplitude is equal to zero at the fiber input. Then, the LPFG transmission coefficient for one cladding mode has the form
(24)$$T=1-\frac{\kappa^{2}}{\eta^{2}}\sin^{2}\eta L.$$

This function has a dip shape with sidelobes. The depth at the center (δ=0) is equal to $\sin^{2}\kappa L$, and the minimum of a transmission is reached at κL=π/2.

## 3. Fabrication and Characterization of Long-Period Fiber Gratings

### 3.1. LPFGs Fabrication

In general, commercial fusion splicer machines have been adapted in order to inscribe long-period fiber gratings. Typically, they produce an AC arc discharge, the known exception being the one used by Rego et al., the BICC AFS3100 fusion splicer [64]. Some authors use no pulling tension [77,81,92,93,94] while others use some tension to keep the fiber straight and in the same relative position in-between the electrodes [35,37,64,70,74,95,96,97]. Over the years, only a few research groups developed their own high-voltage AC power supplies and all have used an external pulling tension in order to inscribe the gratings [98,99,100]. Generally speaking, the fabrication setup consists of a white light source to inject the light into the fiber, an optical spectrum analyzer to register the grating transmission spectrum, a motorized translation stage to move the fiber, and an arc discharge that is applied periodically to the fiber without coating. The fiber can be under strain, and a weight of 1 g to 40 g is used to keep it straight in-between the electrodes. Basically, an arc-discharge is applied to the fiber with an electric current ranging from 5 to 20 mA and a duration of 200 ms to 2 s, and the fiber is then moved by the grating period and another arc discharge is applied. The amount of energy transmitted to the fiber depends on the electric current and voltage and also on the arc duration and, therefore, different combinations of the fabrication parameters can be used. The transmission spectrum is monitored in real time and the automated process arc discharge/fiber displacement is stopped when a resonance with the desired amplitude dip at a specific wavelength is reached. In general, the grating length is shorter than 3 cm (period of 540 μm and 40 arc discharges to reach more than −20 dB amplitude dip), and it takes less than 5 min to produce it. In recent years, G. Yin et al. [101] adapted the commercial Fitel fusion splicer FSM-100 in order to produce the gratings without the need to use an external high-resolution translation stage. The system is compact and robust, and the produced gratings are reproducible. The drawbacks are the cost of the fusion splicer, the limited length of the gratings produced and, more relevantly, the minimum grating period achievable that prevents the fabrication in the dispersion turning points, due to the dimensions of the arc discharge.

Over the years we have made several changes to the fabrication setup and we have also developed three high-voltage power supplies. As discussed in [102], we moved from a manual to a fully automated writing process, we surpassed the mechanical instability of the setup by designing a new platform, and with new high-voltage power supplies we tried to increase the flexibility by being able to change the arc-discharge parameters from discharge to discharge. We noticed that by using the AC electric current, we were able to decrease the dimensions of the arc discharge to what would be useful to write the gratings in the dispersion turning points and we also produced several optical devices with this arrangement [22,84]. However, the arc discharge was unstable, not allowing a good reproducibility of the technique. Therefore, we used the commercial BICC AFS3100 fusion splicer as the source of the stable-DC high-voltage power supply and the new platform with mechanical stability and with the desired flexibility that enables the fabrication of reproducible gratings [103]. Figure 2 shows the evolution of the setup arrangement in order to achieve mechanical and electrical stability.

Besides the high-voltage power supply, the other key point is the central block we developed that comprises two independent v-grooves set and guarantees the position of the fiber in-between the electrodes. The first pair of v-grooves allows for a coarse alignment and the second, made in a single block, it is located closer to the electrodes and guarantees the relative position between the fiber and the arc discharge. The position of the electrodes can be adjusted in the plane perpendicular to the fiber (see Figure 3). We also developed our own setup for polishing the electrodes after normal degradation caused by the arc discharges [102]. It should be highlighted that the final experimental setup shown in Figure 2d), where the high voltage is instead supplied by the BICC fusion splicer, has been used by dozens of students in their final BSc projects and MSc/PhD thesis as well as by researchers worldwide, resulting in more than two hundred published papers [86,104,105,106,107,108,109,110,111,112]. The only drawback of this infrastructure is the limitation in the minimum grating period to be used, of about 385 μm in standard fibers (240 μm in a nitrogen doped fiber [82]).

### 3.2. LPFGs Characterization

To shed some light on the mechanisms of gratings formation, it is important to understand how the type of arc discharge, the electric current and time duration, the type of fiber, the external tension applied, the position of the fiber in-between the electrodes, and the fiber temperature will affect the type of grating produced. In general, commercial fiber fusion splicers are used to fabricate the LPFGs. The electrodes gap typically range from 1 to 3 mm; for fiber fusion, a larger gap is recommended in order to increase the width of the AC discharge, promoting a uniform temperature in the splice region. This is also useful to fabricate reproducible symmetric LPFGs. However, it also imposes a limit on the shortest period to be used due to the thermal diffusion of neighbor arc discharges. We also demonstrated that the type of high voltage power supply used, DC or AC, has implications on the characteristics of the arc-discharge. In particular, we investigated the effect that changing the relative position of the fiber in-between the electrodes has on the reproducibility of the technique and also on type of grating produced. In fact, it was shown that if the fiber is well centered in-between the electrodes, the average temperature is higher and the temperature gradient is low, contributing to the fabrication of symmetric gratings. On the other hand, by moving the fiber to a region with lower average temperature and higher temperature gradient, we promote the fabrication of asymmetric gratings and we increase the reproducibility of the technique. Thus, if one cannot guarantee that the fiber remains constantly in the same relative position inside the arc region and/or guarantee the stability of the discharges, both will contribute to lack of reproducibility of the produced gratings. It should be noted that electrode degradation and changes of the environment conditions, namely, temperature and humidity, will also contribute to the lack of reproducibility.

There are three other fabrication parameters that can be changed in order to optimize the gratings spectrum: electric current, arc duration, and external pulling tension. The increase of the electric current leads to a higher temperature and therefore to an increase of the coupling coefficient, that is, for the same number of arc-discharges applied, we have deeper resonances. Depending on the grating period and on the formation mechanism, longer arc duration may also contribute to an increase of the dips. If an external pulling tension is applied to keep the fiber aligned in-between the electrodes, the increase of the other two fabrication parameters will contribute more strongly to the grating formation due to the decrease in viscosity and consequent fiber tapering. The influence of the fabrication parameters on the LPFGs spectra (Λ = 540 μm) is depicted in Figure 4, by using DC arc-discharges from BICC AFS3100 fusion splicer machine. As it can be seen, the increase of arc duration and electric current lead to a shift towards shorter wavelengths and to an increase of the dips. The increase of the pulling tension also increases the dips and leads to a shift towards longer wavelengths. The reasons for that behavior will be explained in the next section dedicated to the study of the mechanisms of gratings formation. The evolution of the gratings spectra for two different external pulling tensions is depicted in Figure 5a,b. Note that in general the resonance wavelengths remain the same during the grating inscription. However, as mentioned before, the increase of the external tension leads to larger resonance wavelengths. It is also known that recoupling occurs, that is, for a particular resonance wavelength the strength of the dip will increase and then decrease periodically as a function of the number of discharges.

In 2015, headed by the intention of writing gratings in the dispersion turning points, we developed a new AC high-voltage power supply and, by using a 0.9 mm electrodes gap, we were able to accomplish the fabrication of long-period gratings achieving a grating period of 148 μm (see Figure 6). This is, to the best of our knowledge, the minimum grating period attained by this technique [98]. Figure 7 shows the transmission spectra of a 148 μm-grating, in the dispersion turning points, when in air and immersed in water. The grating was written in a B/Ge co-doped fiber, PS 1250/1500 from FiberCore, using 122 arc discharges of 12.7 mA during 680 ms, being the fiber under strain caused by a mass of 2 g.

## 4. Mechanisms of Gratings Formation

### 4.1. Fiber Temperature

In order to properly discuss the mechanisms of gratings formation, it is essential to know the temperature attained by a fiber during an arc discharge. That goal was achieved by using electrically insulated thermocouples [113,114]. Thus, we fabricated Pt and Pt/Rh thermocouples, with different diameters, inside a silica capillary having an interior diameter of 56 μm and an exterior diameter of 125 μm [102]. Afterwards, we applied arc discharges around the thermocouple junction, with typical values used for the gratings fabrication, and we acquired the temperature gradient for that particular sensor configuration. By following heat transfer equations, we estimated the peak temperature reached by a fiber during an arc discharge of 9 mA over 1 s. A value of 1350 °C was obtained with a slope of 60 °C/mA. It was also shown that the fiber reached thermal equilibrium in less than half a second. This information is very important for writing LPFGs in the dispersion turning points since it is possible to introduce large index modulations mitigating the effect of thermal diffusion over neighbor regions affected by the arc discharges. Furthermore, we reached a similar peak temperature by using a different approach based on the blackbody radiation emitted by the fiber [115]. For the common fabrication parameters used to induce the gratings, this methodology pointed towards a value of about 1400 °C, which is very close to the previous one.

### 4.2. Elastic and Viscoelastic Stresses

Diffusion of the core dopants was one of the first mechanisms proposed for the formation of the gratings [37,38]. However, for the typical fabrication parameters used and considering the most common fiber composition, the diffusion length is too short, resulting in a negligible contribution [102,116]. Another potential mechanism is stress relaxation [94,117]. During fiber drawing, due to the different compositions of the core and cladding, the viscosity and the thermal expansion coefficients will be different and, therefore, elastic stresses will be induced. On the other hand, during LPFGs inscription using arc discharges, the temperature is raised above 1000 °C for a short period, allowing for the annealing of those stresses. Systematic stress measurements were performed in gratings arc-induced in N-doped fibers drawn at different tensions. Gratings were inscribed using 40 arc-discharges with a current of 9 mA and 1 s duration separated by 400 μm. No obvious relation between gratings strength or resonance wavelength as a function of the drawing tension could be extracted [118]. On the other hand, it was possible to conclude that the increase of the pulling tension leads to stronger gratings. The one-dimensional stress profiles of the three fibers are shown in Figure 8. A linear dependence of the core and cladding stress on the drawing tension was found. The inset shows the 2D tomographic stress profile of the N-doped fiber drawn with a tension of 125 g.

The effect of the arc discharge on the stress profile of the N-doped fibers is shown in Figure 9. It can be observed that the stress profile changed considerably after the discharge; in particular, stress was considerably annealed in the region submitted to the arc discharge. In the region between discharges the stress profile was also partially annealed. The core refractive index changed due to stress annealing was one order of magnitude smaller than that required to justify the strength of the resonances obtained. It should be stressed that the general picture was obtained for gratings written in all different fibers. As can be observed, the stress profile became asymmetric, being more evident in the region outside the discharge. The increase of the external pulling tension prevented the complete stress annealing and, in the region outside the arc discharge, for the highest tension of 36.3 g, stress was in fact induced. The length of the annealed region was estimated to be approximately 1 mm, which is much larger than the grating period. Therefore, only a weak modulation of the refractive index can be taken into account due to stress relaxation. On the other hand, it is expected that the contribution of stress relaxation could increase as the grating period also increases. A systematic study performed on the SMF28 fiber indicated that, for a grating period of 600 μm (considering a 7 × 10^−5^ index modulation due to stress relaxation), a strong grating can be obtained if the length is of about 27 cm, an order of magnitude higher than typical grating lengths [119].

During fiber drawing, depending on the cooling rates, the glass matrix does not have enough time to reach thermal equilibrium and, therefore, the fiber contains some frozen-in viscoelastic stresses. The temperature at which the glass structure is frozen is called the fictive temperature. If the fiber is annealed for some time at high temperatures, close to the fictive temperature, the glass matrix will be rearranged and the fictive temperature will change; thus, the glass properties such as viscosity, thermal expansion, and refractive index will also change. The annealing of viscoelastic stresses due to the arc discharges has been proposed as a mechanism for gratings formation [120]. In order to relax residual stresses [118,121], the fiber drawn with the highest tension was pre-annealed in air for 30 min at 1050 °C. Note that these annealing conditions not only completely relax residual stresses, but also partially relax viscoelastic stresses [122]. Afterwards, we measured its refractive index profile (Figure 10) [123]. The cladding refractive index increased by 1.5 × 10^−3^, which is one order of magnitude higher than what could be explained by simply considering elastic stress relaxations (Δn = −6.35 × 10^−6^ × Δσ, being Δσ in MPa) (Figure 8). Stress measurements showed that the stresses were completely annealed in the cladding region whilst the core changed from compressive to tensile with a correspondent change of about 60 MPa.

On the other hand, a LPFG written in the pre-annealed fiber (drawn with the 195 g tension) had a coupling strength 3–4 times higher than for the grating written in the pristine fiber despite its length being only half of the latter. Figure 11 also shows that the resonance dips shifted towards shorter wavelengths compared with the grating induced in the pristine fiber. This may be attributed to the annealing of viscoelastic stresses in the cladding as observed in the RIP of the fiber. It should be noted, however, that strong stresses were also observed in the core and inner cladding regions (Inset Figure 11).

Figure 12 shows that the arc discharge induced strong stresses particularly in the core region of the pre-annealed fiber, having the potential to change the core refractive index by 9.5 × 10^−4^. The cladding region was less affected although the region outside the arc discharge changed considerably. Therefore, this may be an indication that the highest heating/cooling rates involved in an arc discharge associated with the use of pulling tensions promote structural changes in the glass matrix [64,124].

### 4.3. Microdeformations

As discussed above, the increase of the external pulling tension increases the coupling strength and, therefore, it is expected that fiber tapering may contribute to gratings formation. However, simulations demonstrated that strong tapers are required, almost 20% modulation, in order to have 100% coupling for symmetric cladding modes (LP_05_) [102]. The higher the order of the cladding modes, the stronger the coupling constant. The average reduction of the fiber diameter leads to a shift of the resonances towards shorter wavelengths. On the other hand, microbending is a well-known mechanism to induce mechanical gratings, being the coupling to asymmetric cladding modes [51,59,125,126,127,128,129]. Therefore, we suspected that arc-induced gratings produced using our setup would also be asymmetric. The performed simulations demonstrated a better fitting for asymmetric gratings in the case of the SMF28 fiber. In contrast, for the B/Ge doped fiber, the best fitting was for symmetric gratings. The measurement of the refracted near field corroborated the simulations (Figure 13) [129].

Afterwards, by analyzing the type of arc discharge, it being directional, we found out that a temperature gradient exists and that was considered the reason for the origin of the microdeformation induced in the fiber core. Thus, the fiber tapering was asymmetric, which led to the microdeformation that was indeed responsible for the formation of the grating (Figure 14) [103]. It is interesting to note that, in the case of asymmetric cladding modes, the coupling constant reaches 100%, first for LP_12_ followed by LP_13_ and finally LP_14_ and LP_11_ cladding modes. This was also observed during gratings fabrication by following the spectra evolution (Figure 15).

Concerning the B/Ge codoped fiber, it was possible, by moving the fiber to a region of the arc with a lower average temperature and a higher temperature gradient, to induce two superimposed gratings originated by different mechanisms (Figure 16). The first created resonances at longer wavelengths and they resulted from coupling to symmetric cladding modes. At shorter wavelengths, the resonances belonged to an asymmetric grating (Figure 17). The symmetric grating was annealed by heating the fiber for 30 min at 800 °C and, therefore, the formation mechanism was attributed to densification caused by viscoelastic stress relaxation [130]. The asymmetric grating had the general behavior of LPFGs produced by microdeformations. It was also demonstrated that this position of the fiber in-between the arc-discharge leads to the fabrication of reproducible gratings.

Figure 17 shows the dispersion curves for gratings produced by symmetric and asymmetric perturbations. It is observed that resonances belonging to the symmetric cladding modes had dips at longer wavelengths [102].

As mentioned above, mechanically induced grating is a well-known way to couple the core mode to asymmetric cladding modes due to the intrinsic microbending associated with the technique [131]. It should be stressed that these gratings are very helpful in research labs since they are easy to implement, low cost, and the perturbation in the fiber is transient/reversible and, thus, it can be used to estimate the position of the resonance wavelengths in a specific fiber and for demonstration of proof of concept in applications involving LPFGs in optical communications and sensing domains [132,133]. Therefore, it is imperative to compare both arc- and mechanically-induced gratings. Figure 18 shows the dispersion curves for both types of gratings and it is observed that the resonance wavelengths of arc-induced gratings were shorter than the resonances belonging to the mechanical-induced ones. Thus, the shift of the resonances was caused by the arc discharge itself. When the electric current or the arc duration increased, the shift towards shorter wavelengths increased, which may be related to the relaxation of residual and viscoelastic stresses as observed namely by the increase of the refractive index of the fiber cladding (Figure 8, Figure 9 and Figure 10). Moreover, a decrease of the core refractive index was also measured for LPFGs inscribed in the SMF28 fiber with no external tension [119]. On the other hand, the increase of the external pulling tension led to stronger fiber tapering (a decrease of the core and cladding radius) and, therefore, since those changes affect more significantly the effective refractive index of the core than that of cladding [103], from Equation (23) the shifts should also be to shorter wavelengths. Nevertheless, Figure 4 shows precisely the opposite displacement of the resonance peaks. The behavior can be explained by the fact that the external pulling tension prevents stress relaxation and indeed it can also create new stresses in the core and cladding (Figure 9), leading to the observed shift. Finally, concerning symmetric gratings in standard fibers, such as the case of the SMF28 fiber, when no pulling tension is used and the thermal gradient due to the arc discharge is negligible, that is, the fiber is well centered in between the AC discharge, the mechanisms may be a relaxation of elastic/viscoelastic stresses.

## 5. Conclusions

We reviewed the most important achievements of our research group at INESC TEC related to the fabrication of long-period fiber gratings using the electric arc technique. We demonstrated the possibility to virtually fabricate, in a reproducible way, gratings in any kind of fiber. The development of a homemade high-voltage power supply enabled us to inscribe gratings in the dispersion turning points, with a minimum grating period of 148 μm. The influence of the fabrication parameters on the gratings spectra was analyzed. The mechanisms responsible for the formation of the gratings were discussed, based on the measurement of the temperature reached by the fiber during an arc discharge. Depending on the fiber, on the arc discharge, and on the relative position of the fiber/arc-discharge it was possible to simultaneously induce two gratings of different symmetries: coupling of the core mode to symmetric and/or asymmetric cladding modes. In this context, the mechanisms of densification and microdeformations were discussed, as well as the influence of the fabrication parameters on them.

## Figures and Tables

**Figure 1 sensors-21-04914-f001:**
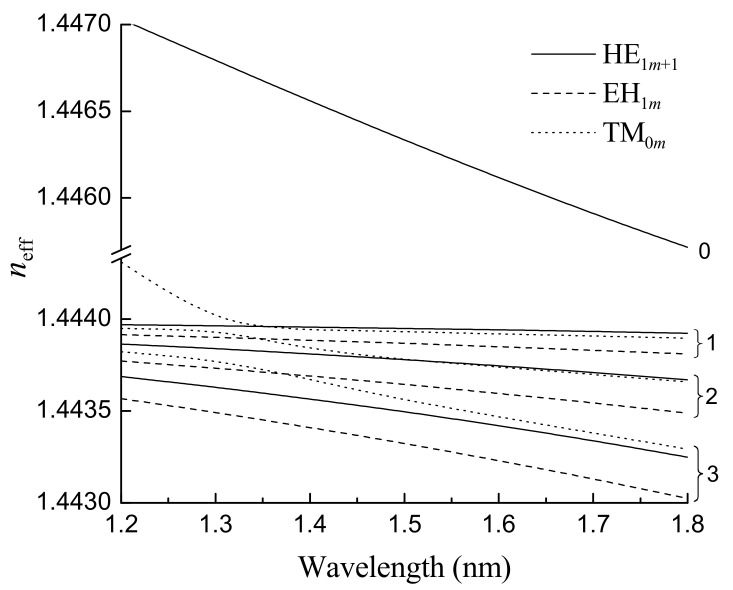
Dependences effective RI of the core mode and some cladding modes on wavelength. Numbering of the curves indicate the radial modal number *m*. Reprinted from [89].

**Figure 2 sensors-21-04914-f002:**
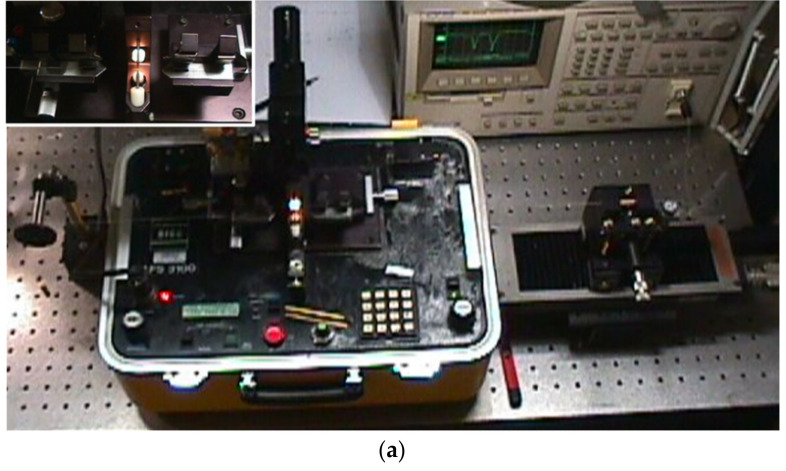
Evolution of the experimental setup for LPFGs fabrication. (**a**) Setup by O. Okhotnikov comprising three independent blocks: the BICC fusion splicer, a motorized translation stage, and a pulley. Inset: v-grooves and electrodes of the fusion splicer; (**b**) Improvement of the alignment of the previous setup and computer-controlled arc discharge (the arc parameters had to be previously defined); (**c**) Setup by F. M. Araújo comprising the homemade high-voltage power supply (unstable), the electrodes support, and the fiber alignment block. Inset: The central block allowing for the positioning of the fiber in-between the electrodes (poor alignment confirmed by using the microscope); (**d**) New setup comprising another high-voltage power supply (unstable) and mechanical stability achieved by the use of three independent blocks: the motorized translational stage, the pulley and the central block with two independent v-grooves set, and the electrodes support. Adapted from [102].

**Figure 3 sensors-21-04914-f003:**
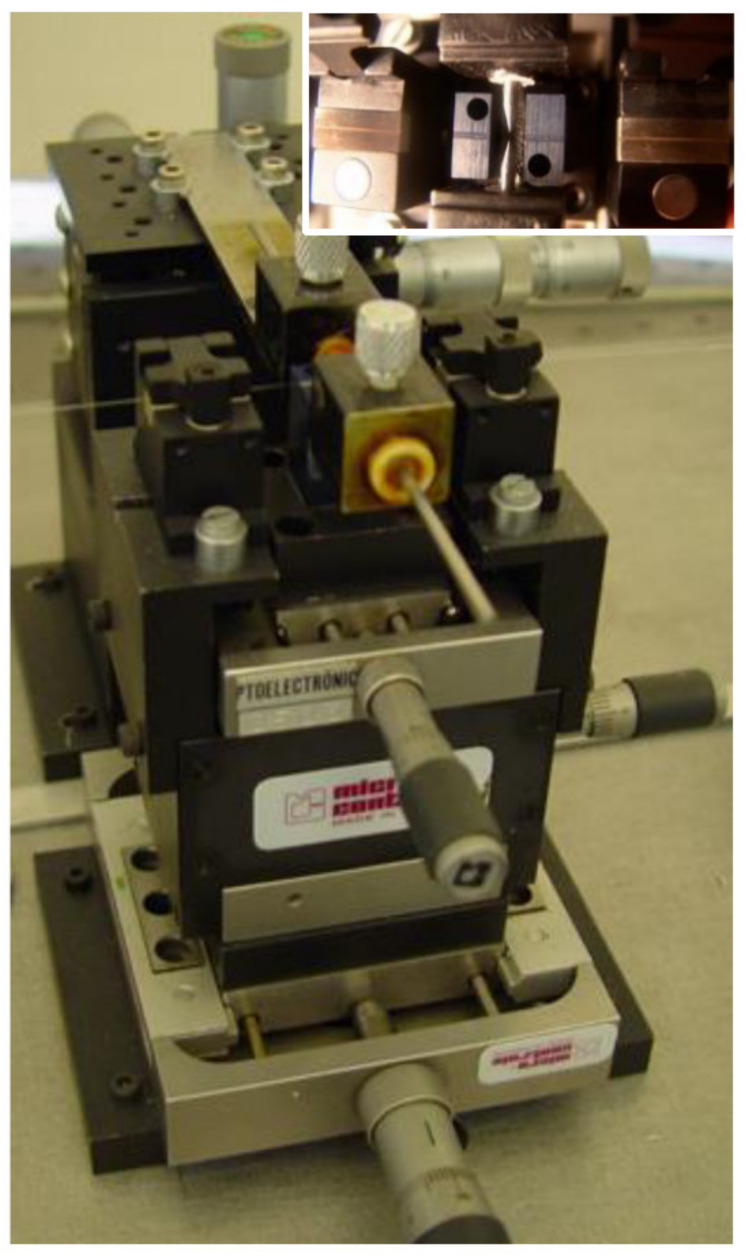
Central block showing the electrodes’ support insulated with Teflon and the v-grooves mounted in different positioners. Inset: closer view of the electrodes and the v-grooves for fiber alignment.

**Figure 4 sensors-21-04914-f004:**
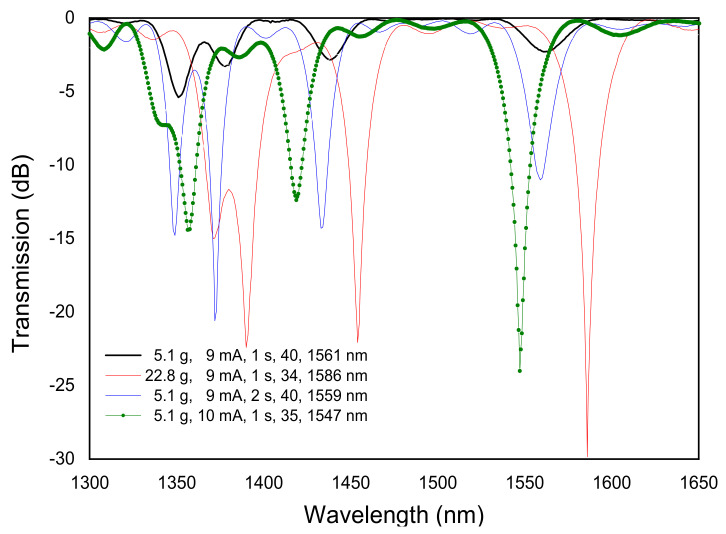
Influence of the fabrication parameters on LPFGs arc-induced in the SMF28 fiber. Adapted with permission from ref. [82]. © 2005 Taylor & Francis.

**Figure 5 sensors-21-04914-f005:**
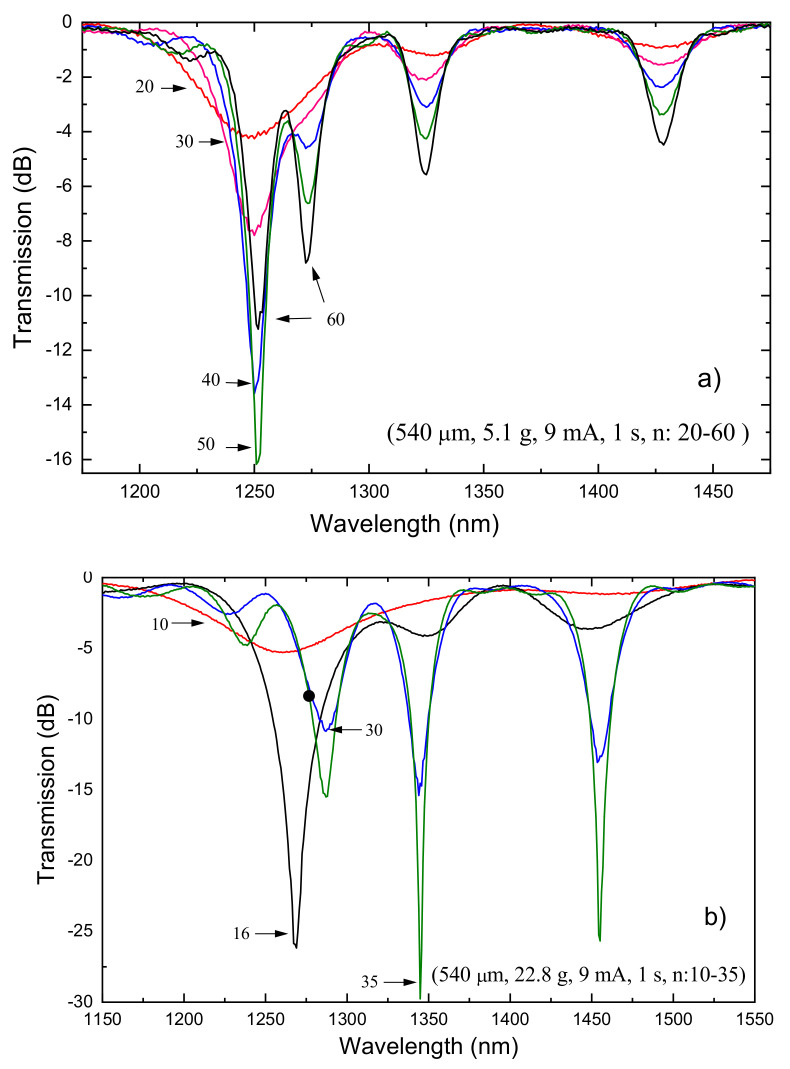
Influence of the fabrication parameters. Evolution of two LPFGs inscribed in the Sumitomo fiber (1.5 mol% GeO_2_) using an external tension of (**a**) 5.1 g and (**b**) 22.8 g. Reprinted from [102].

**Figure 6 sensors-21-04914-f006:**
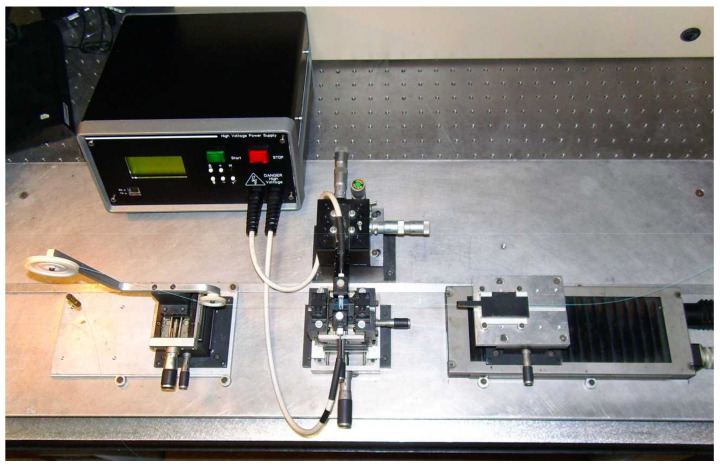
Experimental setup for LPFGs fabrication in the dispersion turning points. Reprinted with permission from ref. [98]. © 2016 IEEE.

**Figure 7 sensors-21-04914-f007:**
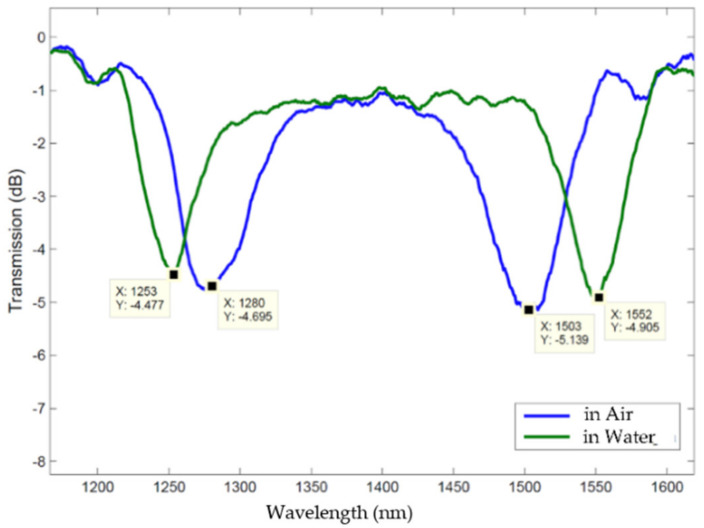
A 148 μm-LPFG in the DTP, arc-induced in a B/Ge co-doped fiber, immersed in air/water.

**Figure 8 sensors-21-04914-f008:**
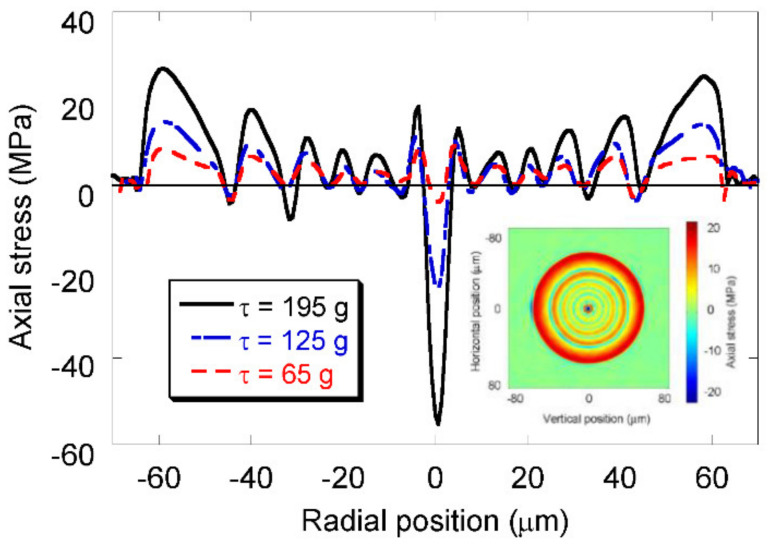
One-dimension tomographic stress profiles of the three fibers. Inset: 2D tomographic stress profile of the fiber drawn with a 125 g tension. Adapted with permission from ref. [118]. © 2005 IEEE.

**Figure 9 sensors-21-04914-f009:**
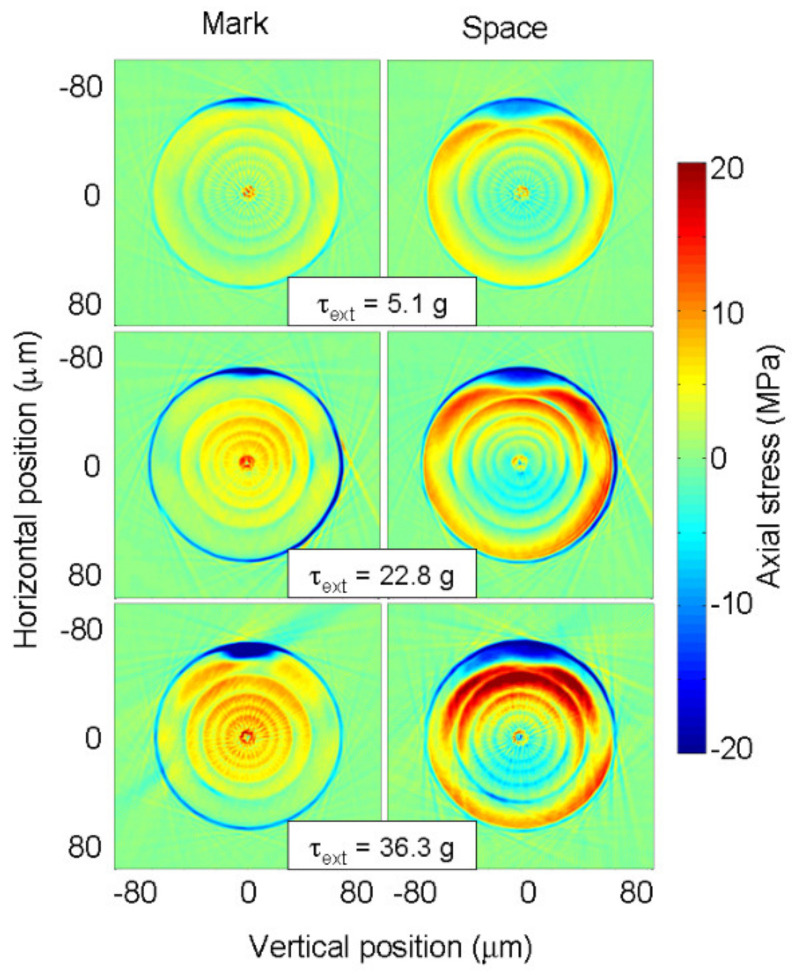
Tomographic stress profiles of the fiber drawn with the highest tension where LPFGs were induced using different pulling tensions. Reprinted with permission from ref. [118]. © 2005 IEEE.

**Figure 10 sensors-21-04914-f010:**
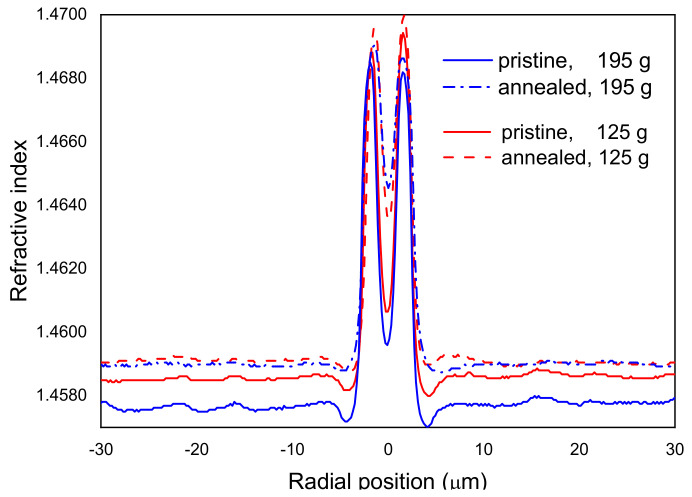
Refractive index profiles of the fibers drawn with 125 g and 195 g, before and after annealing. Reprinted with permission from ref. [123]. © 2006 IEE.

**Figure 11 sensors-21-04914-f011:**
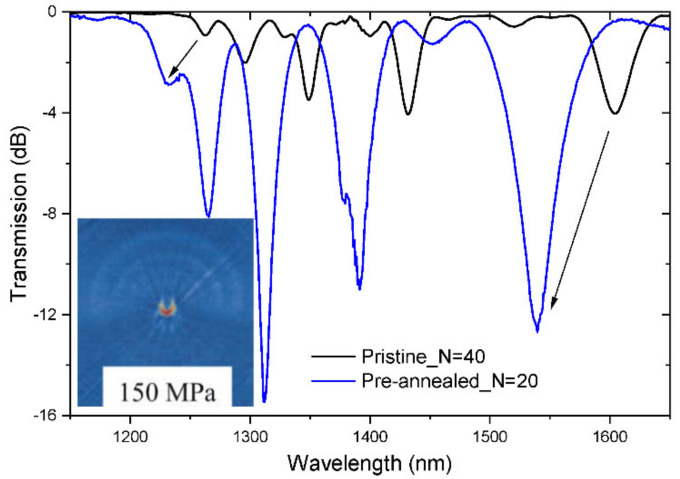
Spectra of gratings written in the pristine and pre-annealed fibre. Inset: Tomographic stress profile showing the “half-moon” stress induced in the core and inner cladding region of the pre-annealed by the arc discharge. Adapted with permission from ref. [123]. © 2006 IEE.

**Figure 12 sensors-21-04914-f012:**
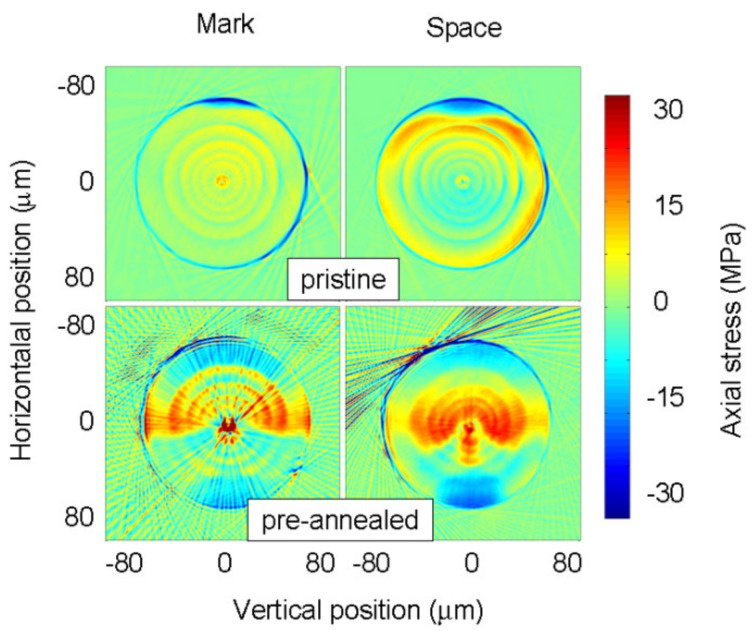
Comparison of the tomographic stress profiles for LPFGs written in the pristine and pre-annealed fibre. Reprinted with permission from ref. [123]. © 2006 IEE.

**Figure 13 sensors-21-04914-f013:**
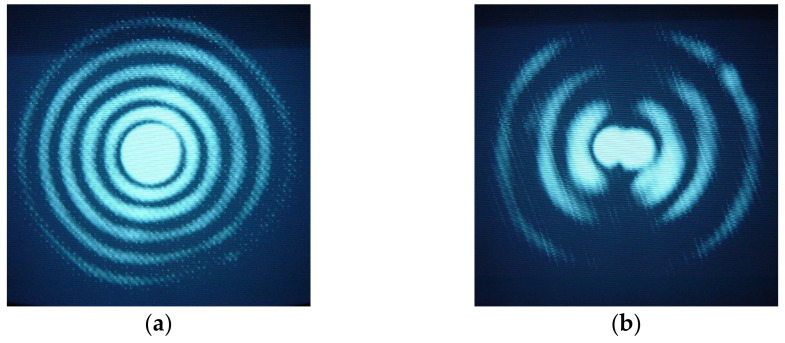
Intensity distribution in the near field for the (**a**) LP_07_ cladding mode of a 415 μm-LPFG written in the B/Ge fiber and (**b**) LP_14_ cladding mode of a 540 μm-LPFG written in the SMF28 fiber. Reprinted with permission from ref. [129]. © 2006 The Optical Society.

**Figure 14 sensors-21-04914-f014:**
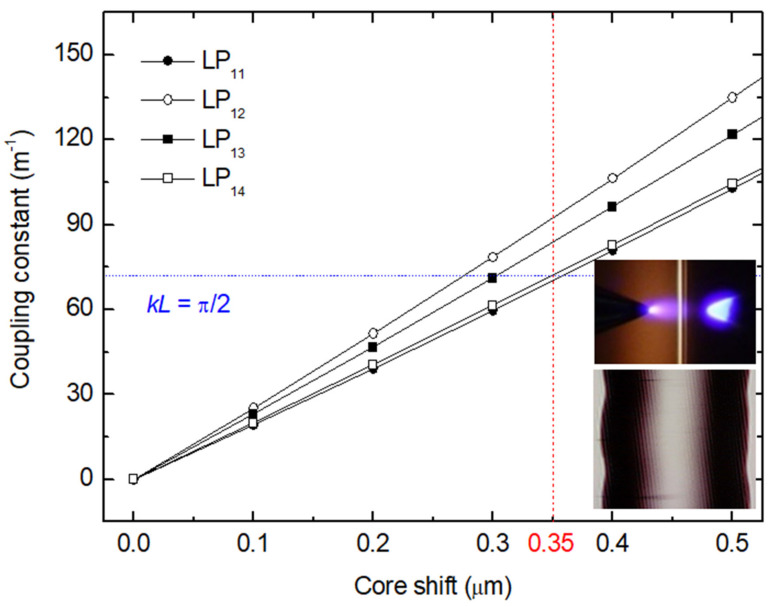
Coupling constant as a function of the core shift. Inset: Photograph of the microdeformation (fiber compressed on its length, showing two modulations separated by 540 μm) caused by the temperature gradient of the DC arc discharge. Adapted with permission from ref. [103]. © 2007 The Optical Society.

**Figure 15 sensors-21-04914-f015:**
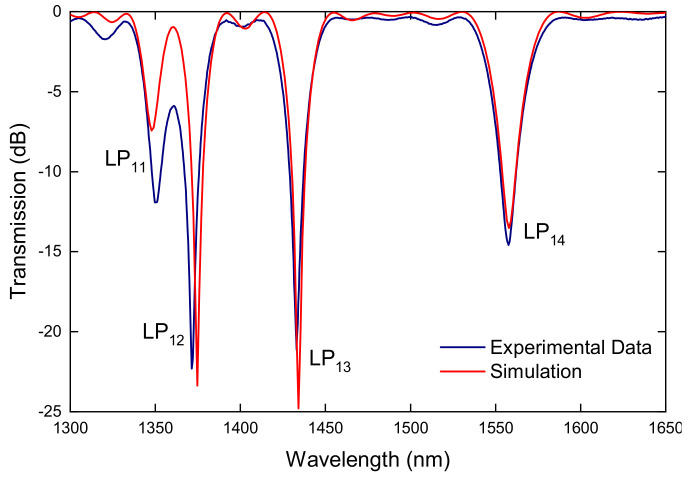
Transmission spectrum of an LPFG: experimental (solid line) and simulation (dashed line). Reprinted with permission from ref. [103]. © 2007 The Optical Society.

**Figure 16 sensors-21-04914-f016:**
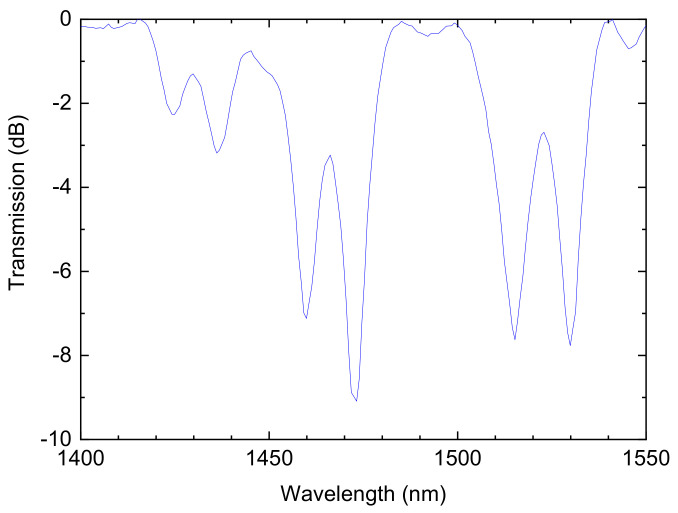
Gratings spectra containing a dual set of resonances inscribed simultaneously in the B/Ge codoped fiber. Reprinted with permission from ref. [130]. © 2007 The Optical Society.

**Figure 17 sensors-21-04914-f017:**
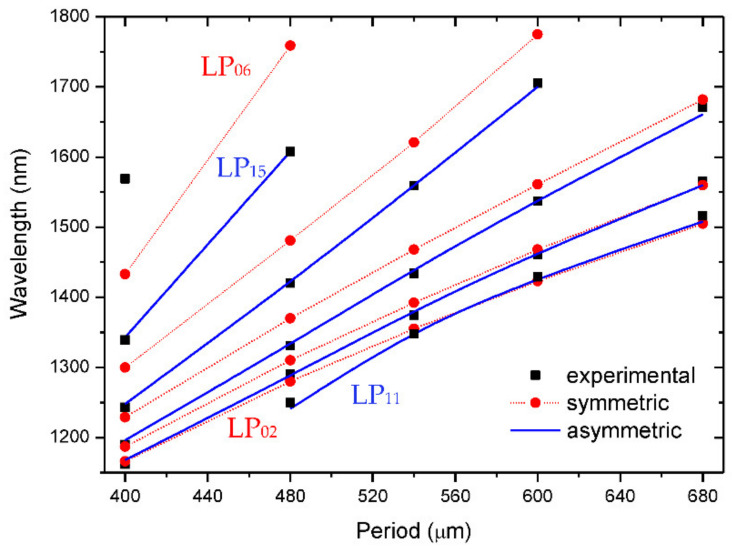
Dispersion curves for symmetric and asymmetric cladding modes. Adapted from [86].

**Figure 18 sensors-21-04914-f018:**
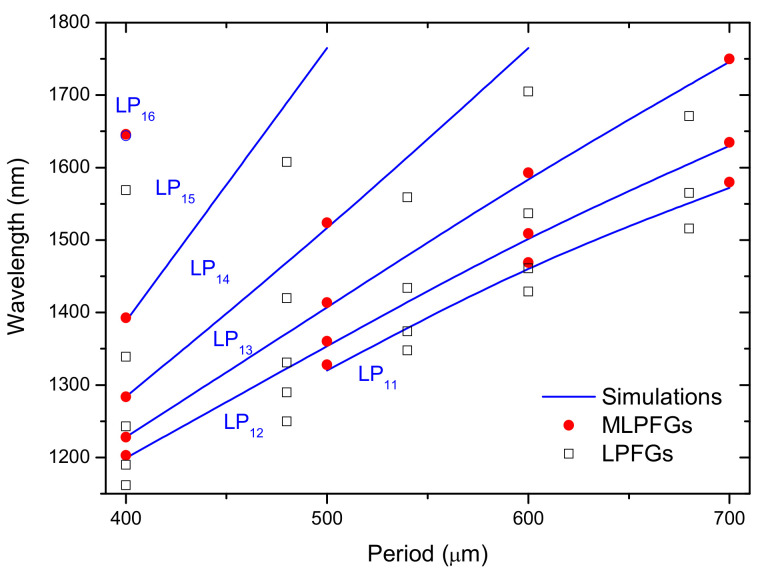
Dispersion curves corresponding to arc- and mechanically-induced gratings in the SMF28 fiber (simulations apply to MLPFGs). Reprinted with permission from ref. [131]. © 2008 Wiley Periodicals, Inc.

## Data Availability

Not applicable.

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
