# Peer review of "Arc-Induced Long-Period Fiber Gratings at INESC TEC. Part I: Fabrication, Characterization and Mechanisms of Formation"

_sensors, 2021, doi:10.3390/s21144914_

Round 1
Reviewer 1 Report
In review this paper, I think this paper write well and it is interesting to the reader. But this paper needs to make minor revision after accepted. The comments are as following.
- In fig. 2, the experimental setup for LPFGs fabrication is not very clear and the resolution of figures is very poor. Could the author give more clearly explanation for the setup and change the new figures.
- In line 454 and 455, the author said that "We have pre-annealed the
fiber, drawn with the highest tension, for 30 min at 1050 ºC ....". Could the author describe the condition of annealing ex. in air or vacuum? Then, please give explanation why choose the temperature and time. - In figure 18, please give more scientific explanation for the difference between the simulation and experimental results.
Reviewer 2 Report
In this work, the authors report a review of the achievements done at INESC TEC Porto regarding the fabrication and characterization of arc-induced long period fiber gratings (LPG).
The paper is fairly written and organized and provides a useful recap for those working in the field of LPGs and especially those fabricated with arc discharge technique.
I have only the following minor comments:
The aim of the paper is clear, however I feel that more comments should be included also regarding what has been done so far also by other groups, as for example the works at Refs. [72-75].
The resolution of Fig. 2 should be improved. I suggest putting same labels to indicate each component of the setups. Moreover, please provide a brief comment to highlight the differences between each setup in (a-d), also in the figure caption.
Different subsection can be created in Sections 3 and 4 to make the reading easier.
